# Pro-Environmental Viticulture: Status Quo and Perspectives from Prosecco Winegrowers in Italy

Elisa Giampietri [1,2] and Samuele Trestini [1,2,*]

1 Department of Land, Environment, Agriculture and Forestry (TESAF), University of Padova, 35020 Padova, Italy
2 Interdepartmental Centre for Research in Viticulture and Enology (CIRVE), University of Padova, Conegliano, 31015 Treviso, Italy
* Correspondence: samuele.trestini@unipd.it; Tel.: +39-049-8272737

**Abstract:** In the last few decades, criticisms arose in society over sustainability in viticulture, due particularly to environmental and health concerns about pesticide use. The presence of social conflicts is well documented in some renowned wine areas as the Bourgogne in France and the Prosecco in Italy. As a novel contribution, this paper provides empirical insights into winegrowers' commitment and will and related motivations towards environmental sustainability in the Prosecco Hills area, where social conflicts are well documented around this relevant facet, but little is known on the part of producers. This study aims to explore the pro-environmental behavioral intention of Prosecco winegrowers, focusing on its behavioral determinants, namely knowledge, responsibility, and self-identity. Data collection was held from August to December 2021 through a structured online questionnaire sent to Prosecco winegrowers. We obtained 87 completed questionnaires and data were analyzed through Partial Least Square Structural Equation modeling using SmartPLS software. The results suggest that, on average, winegrowers in our sample show a high pro-environmental behavioral intention. Moreover, the results associate a higher intention to adopt pro-environmental behavior and a higher responsibility towards sustainable viticulture with winegrowers' having a good knowledge of farming practices beneficial for the environment. Moreover, winegrowers who feel more responsible for acting sustainability towards the environmental in the area and those who view themselves as pro-environmental farmers intend to farm more sustainably. Our results have implications to support the design of locally adapted strategies and policies aimed at improving the diffusion of more sustainable farming practices and resolving local conflicts.

**Keywords:** environmentally sustainable viticulture; pesticide use; behavioral factors; Prosecco; PLS-SEM

## 1. Introduction

In recent years, the issue related to the environmental and health effects of the intense use or misuse of pesticides (i.e., herbicides, fungicides, insecticides) in agriculture has been central in the public debate. In line with this, sustainability in agriculture and, more specifically, reducing pesticide use are major issues in several multi-level policies. For instance, the European Union's Farm-to-Fork strategy aims to cut the use of chemical pesticides in half by 2030.

Viticulture shapes several ecological, economic, and cultural systems. For instance, it represents a strategy for the revitalization and the development of rural areas [1], generates new development opportunities and virtuous synergies between sectors (e.g., agriculture and tourism), and provides cultural heritage and ecosystem services (e.g., biodiversity conservation) [2,3]. Nevertheless, viticulture can also be a source of conflict. In line with this, several criticisms arose in society over high-level or uncontrolled pesticide use in grapevine production and its negative environmental and health impacts in the last few decades. Social conflicts are well documented in some renowned wine areas, for

instance in the Bourgogne wine region in France [4,5] and the Prosecco area in Italy [6]. In both contexts, conflicts arose mainly between winegrowers and local communities (i.e., residents) and/or tourists, over concerns regarding: (i) the impact of pesticides' aerial spraying and drift from the vineyard to public and private places (e.g., schools, homes) that in the view of residents compromises the local quality of life and well-being; (ii) the environmental risks linked to water and soil contamination, and the decline in biodiversity; and (iii) landscape transformation.

As for the Prosecco area, negative externalities have been recently defined as the hidden costs or unintended consequences of wine production [6]. On the one hand, residents' point of view has been extensively documented in several recent works. Accordingly, many authors [6–8] reported complaints related to the creation of a repetitive landscape (i.e., monoculture) with several implications on environmental quality loss (e.g., water contamination, loss of biodiversity, soil fertility degradation), resource redistribution issues, and negative impacts of agro-chemical spraying on health. On the other hand, to the best of our knowledge, Prosecco winegrowers' point of view is scarcely investigated at the moment. The lack of knowledge contributes to local people's complaints about a scarce or null commitment of producers in viticulture sustainable management. Nevertheless, it is plausible to assume that sometimes producers' efforts towards sustainable farming practices are invisible to local citizens because of their lack of knowledge or scarce understanding of the environmental benefits of producers' farming practices [9]. Hence, there is a great need to ascertain how much winegrowers already do and, especially, to explore future sustainability perspectives in the Prosecco area.

To fill this gap, this paper builds on previous literature and explores winegrowers' decision-making with respect to their pro-environmental behavioral intention. Indeed, deepening knowledge of the determinants of winemakers' behavior is essential to provide a more comprehensive view of the sustainability issue in the Prosecco area, to support local administrators' policy design and local stakeholders to formulate more effective strategies and manage conflicts in the area. All in all, this paper contributes to the literature on decision-making in viticulture that is currently limited [10].

## 2. Background and Hypotheses Development

### 2.1. Background on the Prosecco Wine Area

Prosecco is an iconic, well-known sparkling wine that is produced in two Italian regions in the North-East, namely Veneto and Friuli-Venezia Giulia. This wine, which is produced from Glera grapes, is currently produced in three denominations of origin, namely Prosecco DOC (i.e., Controlled Denomination of Origin), Conegliano Valdobbiadene DOCG (i.e., Controlled and Guaranteed Denomination of Origin), and Asolo Prosecco DOCG. Its increasing success worldwide [11] has documented the redemption of many winegrowers in this historically marginal area—who previously suffered poverty—also contributing to making the entire area very famous. Accordingly, nowadays Prosecco's hills are famous worldwide for their distinctive beauty, history, traditions, and winegrowing vocation embodied in the production of one of the world's most exported wines. All of these elements contributed to declaring the "Colline del Prosecco di Conegliano e Valdobbiadene" as a UNESCO World Heritage site in 2019 (https://whc.unesco.org/en/list/1571 (accessed on 10 November 2022)).

As above mentioned, this area has undergone numerous economic and territorial transformations over the last century, and its dynamics attracted the attention of multiple local actors with mixed views, also entering the research agenda of scholars from different research fields. In the last decade, tensions arose in this area whose complexity resulted in both social controversies and protests in the rural community (e.g., the so-called "pesticide-free marches") and legal conflicts receiving attention even from the press. For instance, some years ago a documentary focusing on the use of pesticides in the Prosecco wine area was diffused to a large Italian television audience (https://www.rai.it/programmi/report/

inchieste/La-frazione-di-prosecco-82f70b9c-ce75-461d-b98d-cb5b8894fd58.html (accessed on 20 October 2022)), accentuating local protests between farmers and residents.

In this area, although the local population recognizes the wealth brought by viticulture (i.e., job opportunities and wine tourism development), two major debated issues exist: (i) the massive spread of vineyards in the area (i.e., the so-called monoculture problem), favored by the huge demand on the international market; and (ii) the diffused use of pesticides and their impact on human health.

Several authors [6,7,12] documented the rise of social conflicts and practices (i.e., movements) related to the sustainability dilemma of Prosecco-related viticulture. For instance, in their recent paper, Basso and Vettoretto [13] categorized the impacts of the expansion of Prosecco production in two typologies: (i) the local land-use change, which resulted in the colonization of natural, semi-natural, agricultural land, and woodlands [8], and in biodiversity loss and soil erosion; and (ii) the spread of a monoculture, which in some parts dominates the landscape. Moreover, as elsewhere (see for instance [14]), in this area concerns about the effect of spraying products near schools or homes produced several complaints related to the exposure in people living near fields (i.e., residents) [6]. Over the years, these issues have led to the emergence of many citizens' movements. These represent small-medium groups that are often different from each other while sharing a common objective: denouncing social and environmental concerns about the sustainability of vine-growing practices in the Prosecco Hills area and clamoring for change. Parallel to this, various forms of intervention have been initiated to facilitate the dialogue between residents and winegrowers, as those promoted by the local church and the prefect. In addition, several municipalities in the area adopted so-called local police regulations to regulate the use of treatments by winegrowers. In addition to this, the producers' consortia (i.e., the wine industry) have adopted and implemented environmental certifications and schemes to address key sustainability challenges over the last years. Nevertheless, nowadays conflicts and tensions persist, and protests continue in the Prosecco area [6], with the unchanged position of local committees and movements holding firm on the goals and quest for a more sustainable viticulture. On the other hand, the position of winegrowers remains unknown at the moment, in some ways also preventing an easy resolution.

### 2.2. Hypotheses Development

Pro-environmental behavior can be defined as the individual conscious effort to minimize the negative environmental impact of one's own actions [15] (i.e., grape farming practices). In the context of this study, pro-environmental behavior refers to the minimization of the environmental impact of grape production, e.g., through sustainable pest management. The literature shows that the willingness to reduce the use of pesticides by farmers (also winegrowers) is influenced by the intention to reduce negative impacts on both health and the environment; however, the greatest influence lies in providing environmental benefits [16,17]. Following this, although we acknowledge the interest in understanding Prosecco winegrowers' pro-health behavioral intention, we argue that this is not in the scope of this paper. Indeed, in this study, the focus is only on pro-environmental behavioral intention, while health issues are not addressed.

So far, the literature made extensive use of well-known theoretical frameworks to analyze pro-environmental behavioral intention, intending to test their validity in specific geographical farming contexts and/or concerning specific environmental behaviors. For instance, the norm-activation model (NAM) [18] and the theory of planned behavior [19]. Conversely, this study aims to provide relevant stakeholders with some very first insights into whether winegrowers in the Prosecco Hills area intend to care for the environment and what drives their intention. For this reason, testing theories is not the objective of our study, neither is comprehensively mapping all the potential determinants; this paper focuses attention on a few relevant determinants that the literature largely demonstrates to be associated with farmers' intentions.

The literature on the determinants of farmers' pro-environmental behavior and their inter-relationships is wide (for a more complete understanding, see for instance [20,21]). In a seminal paper [22], the authors suggest that both altruism (e.g., preventing negative impacts on other people's health or on the environment) and self-interest (e.g., minimizing one's own negative impacts) are motivators of people's pro-environmental behavior. The authors also argue that the related intention is linked to the interplay of cognitive (e.g., knowledge) and personality variables (e.g., personal responsibility). This confirms the recent scholars' interest in the role of behavioral determinants of farmers' decision-making regarding the adoption of more environmentally sustainable practices [9].

Hence, both farmers' knowledge (or awareness) that performing some farming practices provides benefits for the environment, and their personal responsibility for acting towards environmental sustainability (i.e., the feeling they can reduce negative environmental impacts) influence pro-environmental intention and behavior [20,23]. For instance, many authors identified lack of knowledge as a major barrier to farmers' uptake of environmentally sustainable farming decisions, as for agroforestry system development [24] or non-participation in voluntary agri-environmental schemes [25]. As for winegrowers, Marques et al. [26] found that conventional producers in Spain mentioned a lack of knowledge as an obstacle to adopting cover crops. Moreover, Reimer et al. [27] found that farmers in Indiana with a higher environmental responsibility are more likely to adopt conservation practices. In line with this, Rezaei-Moghaddam et al. [28] found a positive association between responsibility and farmers' adoption of clean technologies (i.e., recycling agricultural residues by composting them). It is worth considering that, as suggested by Pan et al. [29] who cite some other authors, the effect of knowledge can also be indirect, that is, it can influence behavioral intention through responsibility.

Furthermore, the literature advocates the relevance of another motivator of farmers' decision-making with respect to environmental behavior, namely self-identity [20,30,31]. This latter represents the extent to which a specific behavior is considered to be a part of the self [32]. Self-identity represents the internal frame of reference of the farmer (i.e., the value system and worldview, experiences) that determines his/her preferences [20]. For instance, Lokhorst et al. [33] identified self-identity as the major predictor of farmers' intention to perform non-subsidized agri-environmental measures. In line with this, Hyland et al. [23] found that the decision to adopt mitigation measures is associated with the environmental self-identity of farmers. In addition, Valizadeh et al. [34] found a positive and significant effect of self-identity on farmers' intention towards participation in the management and conservation of wetlands in Iran, while Cullen et al. [35]—who cited some other relevant studies—showed self-identity affecting farmer's participation in agri-environment schemes in Ireland.

In addition, the consumer literature (see for instance [36,37] has extensively analyzed the role of self-identity in influencing pro-environmental behavior. However, this is not the subject of the following analysis, which instead focuses on farmers and especially winegrowers.

Given this background, the above-mentioned behavioral determinants may explain winegrowers' intention towards pro-environmental farming in our case study. Based on this, we explore the degree to which, in our sample, winegrowers' pro-environmental farming intention is associated with knowledge, responsibility, and self-identity and, following the literature [29,38], we explore whether knowledge influences responsibility. Therefore, this study proposes a theoretical model that includes a total of four constructs, with both intention and responsibility being endogenous. Indeed, knowledge affects the intention directly but also indirectly, through responsibility. Therefore, the following four hypotheses are postulated:

**H1.** *Winegrower's knowledge positively affects his/her pro-environmental behavioral intention.*

**H2.** *Winegrower's responsibility positively affects his/her pro-environmental behavioral intention.*

**H3.** *Winegrower's self-identity positively affects his/her pro-environmental behavioral intention.*

**H4.** *Winegrower's knowledge positively affects his/her pro-environmental responsibility.*

## 3. Materials and Methods

This study stems from a research project funded by the European LEADER Program, named "Social Innovation for the Sustainable Development of Viticulture in the Alta Marca" (InnoSoSS). This project applied a participatory approach involving local winegrowers from a specific Prosecco area called "Alta Marca Trevigiana", in the Veneto region (Italy). Data collection was held from August to December 2021 through a structured online questionnaire sent by the extension services through their e-mail list and pre-tested on a small sample of winegrowers (N = 10). The pre-test was conducted within the study area, and its numerosity, which is in line with the literature, is justified by the need to test only the winegrowers' full understanding of the questionnaire. Moreover, an initial sample of farms (N = 100) was selected that was geographically representative, based on the area planted with vines (in hectares) in each municipality in the surveyed territory. However, a smaller number of winegrowers participated in the (voluntary) survey. Indeed, we obtained 87 completed questionnaires. Regarding the small sample size, some authors (see for instance [16]) argue that this is common in studies with farmers, especially when investigating sensitive topics as in this case (i.e., on viticulture environmentally sustainable management).

The questionnaire was structured as follows: first, winegrowers were asked to measure their degree of agreement with some 5-point agree/disagree Likert-type items measuring the intention to adopt behavioral actions to promote sustainable viticulture, namely the pro-environmental behavioral intention (INT). In the second section, several 5-point Likert agree/disagree scales were included to measure farmers' responsibility towards sustainable viticulture (RESP; two items), and their self-identity (SELF; two items). In addition, some 5-point Likert-type items (1 = not at all important; 5 = very important) were used to measure the winegrowers' degree of knowledge (KNOW), i.e., the importance they attach to pro-environmental farming practices. The use of a scale to measure knowledge was already adopted by other authors [29]. In this case, the scale measures the importance respondents assign to certain practices in achieving/maintaining environmental sustainability of viticulture. Since all of the items refer to environmental sustainability, the greater the importance winegrowers assign to the items, the greater their knowledge. The items were derived from the literature—as for INT and RESP [29] and for SELF [39]—with adjustments. These latter derived from the answers provided by some local stakeholders in preliminary focus groups, regarding the content of controversies and conflicts in the Prosecco area. Additionally, socio-demographic information on the sample was collected in the third section of the questionnaire.

To analyze the simultaneous relationship between different determinants of winegrowers' intention (INT) to engage in more sustainable viticulture (i.e., reducing its environmental impact), a Partial Least Square Structural Equation Model (PLS-SEM) is estimated through an iterative sequence of OLS regressions, using SmartPLS software. This software is currently increasingly used for similar research as it is an excellent choice for exploratory research and with small samples, as in this case. PLS-SEM is a variance-based procedure well suited for exploratory research [40], even with a limited sample. The estimated path model consists of the outer model (related to the relationship between each construct and its indicators) and the inner model (related to the structural paths between the constructs, both endogenous and exogenous) [41]. Following Hair et al. [40], we combined several collinear indicators (according to a high VIF) into a new composite indicator, i.e., an index that was calculated as the average value of the collinear indicators. This regards the composite indicators for the intention $int_{2\_3}$ (i.e., index from the items $int_2$ and $int_3$) and the knowledge $know_{2\_3\_4}$ (i.e., index from the items $know_2$, $know_3$, and $know_4$). The outer model has been evaluated in terms of indicator reliability (i.e., if the construct explains the variance of each indicator, measured through the standardized loadings), internal consistency reliability (i.e., all the indicators measure the same construct, measured through Cronbach's alpha,

composite reliability $\rho c$, and the Dijkstra–Henseler $\rho a$), convergent validity (i.e., a measure of the amount of variance that is captured by a construct compared to the amount of variance due to measurement error, measured through the average variance extracted—AVE), and discriminant validity (i.e., if an indicator represents only its own construct, measured through the Heterotrait–Monotrait criterion or HTMT, and the Fornell–Larcker criterion). As for the inner model, this was evaluated through predictive validity ($Q^2$) using the blindfolding approach, the explained variance ($R^2$) using bootstrapping, and the estimation of the path coefficients (i.e., standardized β coefficients). Indeed, non-parametric bootstrapping provides a more precise PLS estimation [42] and we used 5000 subsamples for bootstrapping to check significant path coefficients [41].

## 4. Results

Table 1 shows that the investigated sample consists mainly of males (85%), the mean age is 48 years, and 16% had higher education (i.e., a university degree). The main size of the farm amounts to 8 hectares. As for the production method, more than half of the sample adopts voluntary integrated agriculture (54%), while only a minority (5%) is organic. Moreover, 29% of the farms adopt a sustainable certification scheme (e.g., Equalitas, VIVA, etc.). Hence, a large part of the sample already adopts sustainability-related production practices or adheres to certification schemes.

**Table 1.** Sample descriptive statistics (N = 87).

| Variable | Description | Mean | Std. Dev. | Min | Max | % |
|---|---|---|---|---|---|---|
| Age | (years) | 48 | 16 | 20 | 88 | |
| Gender | 1 = male, 0 = woman/other | | | | | 85 |
| University degree | 1 = yes, 0 = otherwise | | | | | 16 |
| Farm size | (hectares) | 8 | 10 | 0 | 60 | |
| Integrated production * | 1 = yes, 0 = otherwise | | | | | 54 |
| Organic production | 1 = yes, 0 = otherwise | | | | | 5 |
| Farm's sustainability certification | 1 = yes, 0 = otherwise | | | | | 29 |

Note: * Sistema Qualità Nazionale Produzione Integrata (SQNPI).

As shown in Table 2, the average score for each indicator is always above the mean value of the relative scale (i.e., three). Interestingly, this suggests that, on average, winegrowers in our sample are prone to pursue environmental sustainability in Prosecco viticulture (i.e., they show a high pro-environmental behavioral intention). Furthermore, they are aware of environmentally sustainable farming practices linked to wine production, they consider themselves sustainable wine producers, and they feel the responsibility to contribute to the environmental sustainability in the area. It follows that there is a need to spread awareness of winegrowers' efforts in terms of viticulture sustainable management among the local community.

To consider the model having an acceptable fit, we refer to cut-off values of the standardized loadings, Cronbach's alpha, composite reliability $\rho c$ and a Dijkstra-Henseler $\rho a$ above 0.7, while above 0.5 for AVE, below 0.9 for HTMT (we refer to this value since many indicators are very similar to each other [43], and above zero for $Q^2$. As shown in Table 3, the standardized loadings for each indicator are higher than 0.7, thus revealing a good indicator reliability, and all the indicators show acceptable values of Cronbach's alpha ranging from 0.69 to 0.78. Indeed, this can be considered adequate [39,44], although 0.7 is commonly considered as the benchmark. Furthermore, composite reliability $\rho c$ and Dijkstra-Henseler $\rho a$ are in the recommended range (0.87–0.90 and 0.69–0.78, respectively), showing internal consistency reliability. Indeed, Mahmud et al. [45]—who cite Dijkstra and Henseler [46]—report acceptable values for the $\rho a$ coefficient above 0.6. Furthermore, all the values for AVE are above 0.5, therefore convergent validity can be confirmed.

**Table 2.** Constructs' indicators and related statistics.

| Indicators | Description | Mean | Std. Dev. |
|---|---|---|---|
| Intention (INT): Referring to your farming activity, please indicate your agreement with the following statements [a]: | | | |
| $int_1$ | I intend to adopt practices and tools that reduce the negative impact on the environment | 4.15 | 0.80 |
| $int_2$ | I will make proper use of agrochemicals to minimize potential environmental problems | 4.43 | 0.82 |
| $int_3$ | I will support the proposals of local institutions to adopt practices and tools for viticulture sustainability | 3.86 | 0.86 |
| Knowledge (KNOW): In relation to wine production, how important do you think the following aspects are for environmental sustainability in viticulture [b]? | | | |
| $know_1$ | Correct use of agrochemicals | 4.57 | 0.76 |
| $know_2$ | Animal and plant biodiversity protection | 4.08 | 0.92 |
| $know_3$ | Preserving soil fertility | 4.43 | 0.74 |
| $know_4$ | Efficient water use | 4.45 | 0.77 |
| Responsibility (RESP): Referring to your farming activity, please indicate your agreement with the following statements [a]: | | | |
| $resp_1$ | As a winegrower, I feel a responsibility to contribute to the protection of the environment | 4.26 | 0.86 |
| $resp_2$ | I feel I have to adopt more sustainable production practices to solve environmental problems | 3.71 | 1.10 |
| Self-identity (SELF-ID): Referring to your farming activity, please indicate your agreement with the following statements [a]: | | | |
| $self\text{-}id_1$ | I consider myself an environmentally sustainable winegrower | 4.08 | 0.81 |
| $self\text{-}id_2$ | I believe I am a very environmentally aware person | 4.09 | 0.83 |

Note: [a] 1 = completely disagree; 5 = completely agree. [b] 1 = not at all important; 5 = very important.

**Table 3.** Outer model evaluation (indicator reliability, internal consistency reliability, convergent validity).

| Construct | Item | Factor Loading | Mean | Std. Dev. |
|---|---|---|---|---|
| INT | $int_1$ $int_{2\_3}$ | 0.907 0.904 | 4.15 | 0.68 |
| KNOW | $know_1$ $know_{2\_3\_4}$ | 0.897 0.892 | 4.38 | 0.67 |
| RESP | $resp_1$ $resp_2$ | 0.899 0.861 | 3.99 | 0.86 |
| SELF-ID | $self\text{-}id_1$ $self\text{-}id_2$ | 0.877 0.869 | 4.09 | 0.72 |

| Construct | Cronbach's alpha | Composite reliability $\rho c$ | Dijkstra-Henseler's $\rho a$ | AVE |
|---|---|---|---|---|
| INT | 0.78 | 0.90 | 0.78 | 0.82 |
| KNOW | 0.75 | 0.89 | 0.75 | 0.80 |
| RESP | 0.71 | 0.87 | 0.72 | 0.78 |
| SELF-ID | 0.69 | 0.87 | 0.69 | 0.76 |

Note: INT = intention; KNOW = knowledge; RESP = responsibility; SELF-ID = self-identity.

As for the discriminant validity, it aims at ensuring that a construct has the strongest relationships with its own indicators in the path model, compared to those belonging to other constructs [40]. Table 4 shows the results for both the Heterotrait-Monotrait ratio [47] and the Fornell-Larcker criterion. Although the former shows some values just above the threshold, Henseler et al. [48] argue that HTMT should be significantly smaller than one to clearly discriminate between two factors. As for the Fornell-Larcker criterion—which has been recently used to confirm discriminant validity by Yang et al. [49]—Table 4 shows that the square root of the AVE (i.e., values on the diagonal) for each construct is higher than its highest correlation (i.e., values in the off-diagonal spaces) with the other constructs, thus confirming discriminant validity. In addition, multicollinearity among constructs is not observed as VIF are ranged 1.38–1.69 [43].

**Table 4.** Outer model evaluation (discriminant validity).

| | Heterotrait-Monotrait Ratio (HTMT) | | | |
| | INT | KNOW | RESP | SELF-ID |
| --- | --- | --- | --- | --- |
| INT | | | | |
| KNOW | 0.765 | | | |
| RESP | 0.910 | 0.640 | | |
| SELF-ID | 0.929 | 0.730 | 0.907 | |
| | Fornell-Larcker Criterion | | | |
| | INT | KNOW | RESP | SELF-ID |
| INT | 0.906 | | | |
| KNOW | 0.585 | 0.895 | | |
| RESP | 0.683 | 0.468 | 0.880 | |
| SELF-ID | 0.682 | 0.524 | 0.638 | 0.873 |

Note: INT = intention; KNOW = knowledge; RESP = responsibility; SELF-ID = self-identity.

Looking at the inner model, Table 5 shows all $Q^2$ values greater than zero, thus confirming good predictive relevance for the model, which indeed effectively reproduces the observed values. Furthermore, the variance of both the intention to behave sustainably and the responsibility is explained at 61% and 22%, respectively, as shown by the $R^2$ values. Our results show all significantly positive effects (Figure 1), hence the willingness to perform a more sustainable grape production likely depends on all the investigated determinants, representing winegrowers' intrinsic factors. Indeed, knowledge ($\beta$ = 0.248), responsibility ($\beta$ = 0.362) and self-identity ($\beta$ = 0.321) are positively associated with the intention to act pro-environmentally. Additionally, the higher the knowledge ($\beta$ = 0.468), the higher the responsibility of winegrowers. Consequently, $H_1$, $H_2$, $H_3$, and $H_4$ are supported.

**Table 5.** Inner model evaluation.

| Hypotheses | | Path Coefficient | *p*-Value | $R^2$ | $Q^2$ |
| --- | --- | --- | --- | --- | --- |
| *H1:* | KNOW => INT | 0.248 | 0.013 | | |
| *H2:* | RESP => INT | 0.362 | 0.001 | 0.611 | 0.478 |
| *H3:* | SELF-ID => INT | 0.321 | 0.008 | | |
| *H4:* | KNOW => RESP | 0.468 | 0.000 | 0.219 | 0.156 |

Note: INT = intention; KNOW = knowledge; RESP = responsibility; SELF-ID = self-identity.

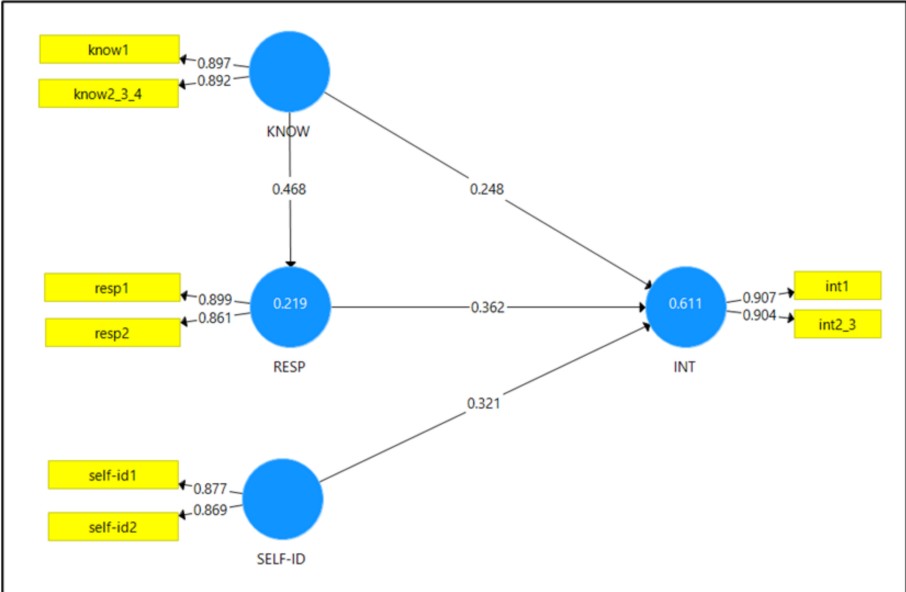

**Figure 1.** Path model with estimated coefficients (INT = intention; KNOW = knowledge; RESP = responsibility; SELF-ID = self-identity).

## 5. Discussion

The study noted that Prosecco winegrowers in our sample show a high pro-environmental behavioral intention, so they are prone to pursue environmental sustainability in the area in the future. Moreover, these winegrowers know well that performing some specific farming practices provides benefits for the environment, they feel responsible for achieving environmental sustainability in the area, and they view themselves as farmers who act pro-environmentally. This novel insight contributes to the recognition of local producers not only as entrepreneurs that are merely interested in generating profit through production but also as environmental caretakers of the Prosecco area.

Moreover, the results from PLS-SEM show the importance of farmers having a strong responsibility towards contributing to the protection of the environment in increasing their intention to act pro-environmentally. Indeed, responsibility is found to be the main predictor of intention, followed by self-identity and knowledge. It follows that winegrowers who feel more responsible for acting sustainably are looking to continue farming pro-environmentally in the future. This positive association between farmers' responsibility and pro-environmental intention is consistent with what was found elsewhere, e.g., by Reimer et al. [27] regarding farmers' adoption of conservation practices in Indiana and Rezaei-Moghaddam et al. [28] towards the adoption of clean technologies in Iran. In line with the literature, this study noted that the higher the winegrower's environmental self-identity, the more he/she is likely to engage in environmentally sustainable farming in the future. Indeed, similar evidence is found in the literature on farmer's pro-environmental behavior worldwide, e.g., regarding water conservation in Iran [50], performing unsubsidized agri-environmental measures in the Netherlands [33,51], participation in wetlands' management and conservation in Iran [34], and participation in agri-environment schemes in Ireland [35]. Zemo and Termansen [52] (p. 333), who cite some other authors, state that "if people act on a certain activity voluntarily without any external pressure, they see the activity as part of the self". Therefore, it is plausible to believe that Prosecco winegrowers who already adopt certifications and/or environmentally friendly agricultural practices perceive a strong environmental self-identity. Since "behaviours associated with self-identity are more likely to persist over time" [20] (p. 286), one possible recommendation is to consolidate the recognition of Prosecco winegrowers' sustainable role, for instance by local Wine Consortia and/or winegrowers' representatives, as it becomes crucial to strengthen their awareness of acting sustainably and thus their self-identity, with a view to promoting sustainability over time. In addition, farmers who are more aware of the importance of specific farming practices for the environmental sustainability of grape production are more likely to sustain local environmental protection. This confirms the evidence of previous works [26] showing that the lack of information may represent a friction to the uptake of more sustainable farming. Another particularly interesting result from the analysis is the evidence that greater knowledge is positively associated with a greater responsibility towards sustainable farming management in the Prosecco area. For instance, a similar effect is documented for the environmental behavioral intention of university students in Taiwan [29]. Hence, improving the knowledge provided to farmers could help in reinforcing the responsibility and, as a consequence, provide a better-suited implementation of more sustainable environmental farming practices in the Prosecco Hills.

## 6. Conclusions

This paper contributes to broadening the knowledge of winegrowers' decision-making. As a novel contribution, it provides empirical insights into winegrowers' commitment and will, and related motivations towards environmental sustainability in the Prosecco area. In that area, social conflicts are well documented around this relevant facet, but little is known about the part of producers. The understanding of winegrowers' decision-making motivators is crucial for tailoring initiatives aimed at raising the environmental performance of viticulture in the investigated area. Following this, our results have implications to support the design of locally adapted strategies and policies aimed at resolving local

conflicts in the area. From a policy perspective, one recommendation is to raise awareness of sustainable farming practices, e.g., through extension or advisory services. Indeed, educating farmers about environmentally sustainable farming (i.e., correct use of agrochemicals, protection of animal and plant biodiversity, preservation of soil fertility, and efficient water use) may increase the adoption of these specific practices. The results also suggest the importance of reinforcing the individual responsibility towards the viticulture environmentally sustainable management, through broadening knowledge and promoting the dialogue with the local community. Indeed, this is a major priority in the area, which could be achieved through the support of Wine Consortia or wine cooperatives, universities, or other relevant actors.

Nevertheless, our findings should be interpreted with caution as these do not apply to the whole population of local winegrowers, and we do not report on the full spectrum of the behavioral determinants of the pro-environmental farming intention. Hence, besides considering a larger and more representative sample, further research is needed to investigate other drivers (e.g., agronomic and psychological factors), also focusing on pro-health behavioral intention, namely the social aspects of sustainability, as those related to the Prosecco-health nexus. The results show that the area's winegrowers manifest a pro-environmental intention, which is positively affected by knowledge, responsibility, and self-identity. This suggests positive prospects for improving the environmental performance of Prosecco Hills' viticulture if appropriate practices (such as fostering greater environmental awareness and information among winegrowers) will be put in place. To conclude, we assume the following study is context specific. However, we hope that future studies can test a similar model in other wine areas experiencing their specific development dynamics, including conflicting ones.

**Author Contributions:** Conceptualization, E.G. and S.T.; methodology, E.G.; investigation, E.G. and S.T.; data curation, E.G.; formal analysis, E.G.; writing—original draft preparation, E.G.; writing—review and editing, E.G. and S.T.; funding acquisition, S.T; supervision, S.T. All authors have read and agreed to the published version of the manuscript.

**Funding:** This research was funded by the research project "Social Innovation for the Sustainable Development of Viticulture in the Alta Marca—InnoSoSS", Veneto region Rural Development Program 2014–2020—measure 16 (LAG Alta Marca)—CUP C24I19002650002.

**Institutional Review Board Statement:** Not applicable.

**Informed Consent Statement:** Informed consent was obtained from all subjects involved in the study.

**Data Availability Statement:** The data presented in this study are available on request from the corresponding author.

**Acknowledgments:** We gratefully acknowledge the winegrowers in the Prosecco area who voluntary participated in the online questionnaire; Alessandro Caputo for helping us with data collection and research assistance; and the extension service, Coldiretti Treviso, that distributed the questionnaire through their own e-mail list. We also appreciated the constructive comments from participants to the XIV International Conference of the European Society for Ecological Economics (ESEE 2022) which was held in Pisa (Italy).

**Conflicts of Interest:** The authors declare no conflict of interest.

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
