# Peer review of "Pro-Environmental Viticulture: Status Quo and Perspectives from Prosecco Winegrowers in Italy"

_sustainability, doi:10.3390/su15021073_

Round 1
Reviewer 1 Report
The article contributes to broadening the knowledge of winegrowers’ decision-making. It provides empirical insights into winegrowers' commitment and will, and related motivations toward environmental sustainability in the Prosecco area. I suggest the to reduce the conclusion part.
Author Response
Thank you for appreciating our work.
Accordingly, we have reduced the conclusions.
Reviewer 2 Report
Dear authors
I read the manuscript entitled “Pro-environmental viticulture: status quo and perspectives from Prosecco winegrowers in Italy”. This manuscript is of an important topic and has focused on a critical issue. It has great potential to be published in Sustainability and I strongly recommend it for publication. However, the authors should try to address the following comments before its consideration for publication.
General comments
1. The share of results should be increased in the abstract section. In other words, please try to mention some more results in the abstract.
2. The main objective of the study should be divided into some operational sub-objectives.
3. The main question/objective and sub-objectives of the study should be added to the manuscript.
4. The discussions and recommendations should be put in an international scope. In other words, please try to go beyond the boundaries of Italy in representation and comparison of the results and discussions.
5. Please try to focus on the broad impacts of the study in the discussion section.
6. The results and discussion section should be presented in two separate sections.
7. Some of the most important limitations of the study should be highlighted in the end of conclusion section.
8. Some more actionable recommendations should be presented for the future research. This part is a necessary part of all conclusions. Please improve this part in your paper.
9. The main take-home message of the study should be presented in the form of one single or two sentences in the end of conclusion section.
Specific comments
1. In page 1, Abstract section, (Lines 11-16), please add some general explanations (type of research, sampling methods, the methods of analysis, etc.) about your research methodology. This is an essential part in an abstract.
2. In page 1, Abstract section, (Lines 18-21), please elaborate the results. The results should be supported using some statistical scores (correlation coefficients, Beta values, indirect effects, and …).
3. In part 2 and (Hypotheses development section) Lines 123-187, please enrich the literature. I have recently read some newly published articles on the relationships of self-identity and responsibility with the pro-environmental behaviors (water conservation, soil conservation, energy conservation, pesticides’ usage, etc.). Please see the following sources and search for more relevant works to enrich this section:
Wang, X., Van der Werff, E., Bouman, T., Harder, M. K., & Steg, L. (2021). I am vs. We are: how biospheric values and environmental identity of individuals and groups can influence pro-environmental behaviour. Frontiers in psychology, 12, 618956.
Carfora, V., Caso, D., Sparks, P., & Conner, M. (2017). Moderating effects of pro-environmental self-identity on pro-environmental intentions and behaviour: A multi-behaviour study. Journal of environmental psychology, 53, 92-99.
Valizadeh, N., Esfandiyari Bayat, S., Bijani, M., Hayati, D., Viira, A. H., Tanaskovik, V., ... & Azadi, H. (2021). Understanding Farmers’ Intention towards the Management and Conservation of Wetlands. Land, 10(8), 860.
4. In page 5, Lines 200-202 the respected authors have mentioned that “Data collection was held from August to December 2021 through a structured online questionnaire sent by the extension services through their e-mail list and pre-tested on a small sample of winegrowers (N=10).” However, the authors should explain that where did they do this pretest study (inside or outside of the study area)? Please also mention why 10 cases you have used in pre-testing phase?
5. In page 5, Lines 202-206, the authors have mentioned that “As a result, we obtained 87 completed questionnaires. Regarding the small sample size, some authors (see for instance [16]) argue that this is common in studies with farmers, especially when investigating sensitive topics as in this case (i.e., on viticulture environmentally sustainable management)”. Please mention the population number in this section. Also, mention why did you collected only 87 completed questionnaires? You should justify the sample size. Also, the sampling approach and process should be justified and explained in more details.
6. In page 5, Lines 2013-2016, the authors have mentioned that “In addition, some 5-point Likert-type items (1=not at all important; 5=very important) were used to measure the winegrowers’ degree of knowledge (KNOW), i.e., the importance they attach to pro-environmental farming practices. The items derived from the literature - as for INT and RESP [29], and SELF [35] - with adjustments.” However, my main question is that how can we measure the knowledge using Likert scale? This should be completely explained and justified. In my opinion, measure knowledge using Likert scale is a bit questionable.
7. In page 5, Lines 221-226, the respected authors have mentioned that “To analyse the simultaneous relationship between different determinants of winegrowers’ intention (INT) to engage in more sustainable viticulture (i.e., reducing its environmental impact), a Partial Least Square Structural Equation Model (PLS-SEM) is estimated through an iterative sequence of OLS regressions, using SmartPLS software. PLS-SEM is a variance-based procedure well suited for exploratory research [36], evenwith a limited sample.” However, the superiorities of this software to the other competing software including AMOS, LISREL, EQS, etc. should be elaborated.
8. Although the authors have mentioned the divergent and convergent validity indices in the results section, this should be summarized in the methods section as well.
9. The results and discussion section should be presented in two separate sections. In present form, it seems that the study has not a discussion section. Of course, I should mention that some of the discussions have been presented withing the contents of “results and discussions” and “conclusions” sections.
10. Results have been elaborated very well.
11. The main limitations and recommendations for the future studies should be explained in the conclusions section.
In general, I believe that this manuscript can be accepted for publication in Sustainability after Minor revisions.
Author Response
We have carefully addressed all the remarks and suggestions in the new version of the manuscript which is a marked-up copy (changes are highlighted in yellow).
In the uploaded file you will find our point-by-point responses to the reviewer’s comments.
